# Exploring the Role of Serum Osteonectin and Hsp27 in Pediatric MAFLD Diagnosis and Cardiometabolic Health

**DOI:** 10.3390/nu16060866

**Published:** 2024-03-16

**Authors:** Anca Bălănescu, Paul-Cristian Bălănescu, Ioana Florentina Codreanu, Iustina-Violeta Stan, Valentina-Daniela Comanici, Alina Maria Robu, Tatiana Ciomârtan

**Affiliations:** 1Faculty of Medicine, “Carol Davila” University of Medicine and Pharmacy, 37 Dionisie Lupu Street, 2nd District, 020021 Bucharest, Romania; anca.balanescu@umfcd.ro (A.B.); ioana.codreanu@umfcd.ro (I.F.C.); iustina.stan@umfcd.ro (I.-V.S.); valentina.comanici@umfcd.ro (V.-D.C.); alina-maria.robu@drd.umfcd.ro (A.M.R.); tatiana.ciomartan@umfcd.ro (T.C.); 2Clinical Research Unit RECIF (Reseau d’Epidemiologie Clinique International Francophone), 19-21 Stefan cel Mare Street, 2nd District, 020125 Bucharest, Romania; 3National Institute for Mother and Child Health “Alessandrescu-Rusescu”, 120 Lacul Tei Boulevard, 2nd District, 020395 Bucharest, Romania

**Keywords:** MAFLD, osteonectin, Hsp27, childhood obesity, cardiometabolic risk indices, lipid-derived indices, VAI, AIP, TG/HDL, TMI, BMI

## Abstract

Background: Childhood obesity is one of the major challenges of public health policies. The problem of fatty liver in childhood, known as MAFLD (metabolic dysfunction-associated fatty liver disease), is of particular interest as the gold standard diagnosis technique is invasive (liver biopsy). Hence, efforts are made to discover more specific biomarkers for the MAFLD signature. Therefore, the aim of the study was to evaluate Osteonectin and Hsp27 as biomarkers for MAFLD diagnosis and to assess their links with auxological and biochemical profiles of overweight and obese pediatric subjects. Methods: A cross-sectional study in which we (re)analyzed data from the MR PONy cohort comprising 71 pediatric subjects. Auxological data, liver ultrasonography and biochemical serum profile were recorded. Lipid-derived indices and body composition indices were calculated. Nevertheless, serum Osteonectin and Hsp27 levels were assessed using an ELISA approach. Results: MAFLD prevalence was 40.8%. Higher Osteonectin levels were noted in MAFLD subjects versus non-MAFLD subjects and in dyslipidemic children regardless of their liver function status. Lipid-derived indices had good diagnostic capacity for MAFLD. Conclusions: We confirm Osteonectin as a MAFLD diagnosis biomarker in children. Also, lipid-derived indices are useful as metabolic-associated organ impairment markers in children even before the onset of obesity.

## 1. Introduction

Obesity remains one of the major challenges of public health policies regarding non-communicable diseases. The childhood obesity-associated burden rises even more concern worldwide due its lifelong cardiometabolic, psychological and even neoplastic complications. Current knowledge speaks about “obesities” in terms of metabolic healthiness. So, the obese phenotype is currently recognized as being polarized in two main categories as follows: metabolically healthy (MHO) and unhealthy (MUHO) obesity [1]. The problem of fatty liver associated with the metabolic disbalances of childhood obesity remains of particular interest. This is currently known as MAFLD (metabolic dysfunction-associated fatty liver disease) or PeFLD type 2 (Pediatric Fatty Liver Disease Type 2)—both being the new and currently used and accepted terminologies for former NAFLD spectrum (non-alcoholic fatty liver disease) [2]. The link between MUHO and MAFLD/PeFLD type 2 remains that of insulin resistance in all age groups.

Efforts are constantly made to discover more specific biomarkers as a signature of MAFLD to help either early diagnosis, better risk stratification, to guide disease monitoring or even as a potential therapeutic target. One promising biomarker (especially for MAFLD progression towards fibrosis) is that of adipokine Osteonectin (SPARC), a matricellular protein [3]. Adult patients with higher Osteonectin expression also expressed higher proapoptotic and profibrogenic markers [4]. As such, one could also consider Osteonectin as a potential biomarker in the evaluation of PeFLD type 2. Nevertheless, obesity (especially MUHO) and MAFLD are highly associated with atherogenesis and eventually cardiovascular risk in adult life. Recently, Hsp27—a heat shock protein under-expressed in atherosclerotic plaques—was confirmed as a biomarker linked to cardiovascular risk in adult patients [5], so the arising question of an existing link between Hsp27 and MAFLD must be addressed.

Therefore, the aim of the study was to evaluate the new candidate biomarkers of adipokine Osteonectin (SPARC) and heat shock protein Hsp27 for PeFLD type 2 (MAFLD) diagnosis and risk stratification. In addition, we aimed to assess the association of their circulatory expression with auxological and biochemical profiles of overweight and obese pediatric subjects. To the best of our knowledge, this is the first study that comparatively evaluates these biomarkers in the pediatric spectrum of MAFLD. Secondarily, the study also aimed to evaluate three cardio-metabolic risk indices (TG/HDL ratio, VAI and AIP) as potential markers to point out MAFLD.

## 2. Materials and Methods

### 2.1. Study Population

We gathered and (re)analyzed data from MR PONy (**m**etabolic and cardiovascular **r**isk factors in a **p**ediatric **o**verweight and obese population with or without **N**AFLD) cohort (already described in our previously published research papers of [6,7]. Briefly, the MR PONy cohort consists of 71 subjects of 3 to 18 years, recruited in a one year time-lapse (January to December 2017) from consecutive pediatric patients admitted to a tertiary pediatric clinic from Bucharest, Romania (INSMC “Alessandrescu Rusescu”). A biobank of serum samples was kept in a −80 Celsius degrees freezer. All patients included were either overweight or obese according to the CDC definition for children [8]. Also, ten lean, healthy children were recruited as part of a control group (mostly to establish the “normality” range of serum SPARC and HSP27 in the pediatric population). None of the subjects had received previous treatment. A parental/legal tutor informed consent signed agreement was mandatory at inclusion. Patients who had secondary causes of obesity (genetic, iatrogenic, endocrinologic), viral infections which could alter liver enzymes (hepatitis C virus, hepatitis B virus, hepatitis A virus, Epstein–Bar virus, Cytomegalovirus) or other chronic diseases affecting liver function (celiac disease, alpha-1 anti-trypsin deficiency, Wilson disease, hypothyroidism) were not enrolled. This cross-sectionally designed study was approved by the local Ethics Committee and was conducted in respect to the Declaration of Helsinki for the ethical principles guiding medical research involving human subjects.

### 2.2. Auxological Data and Derived Indices

Standardized anthropometric measurements according to “WHO STEPS surveillance manual: the WHO STEPwise approach to chronic disease risk factor surveillance/Noncommunicable Diseases and Mental Health, World Health Organization” [9] were recorded at patient inclusion (weight, height, abdominal circumference, midarm circumference). Also, derived indexes were gathered as follows: BMI, TMI, VAI, WtHR.

BMI (Body Mass Index) was calculated as weight (kg) divided by squared height (mp).TMI (Triponderal Mass Index) was calculated as weight (kg) divided by cubed height (m^3^) [10].VAI (visceral adiposity index) [11] as follows:○Male = [WC/(39.68 + (1.88 × BMI))] × (TG/1.03) × (1.31/HDL-C);○Female = [WC/(36.58 + (1.89 × BMI))] × (TG/0.81) × (1.52/HDL-C).AIP (Atherogenic Index of Plasma) = log (TG/HDL Cholesterol) [12].WtHR was calculated as waist circumference (cm) divided by height (cm) multiplied by 100.

All the resulted values from the calculation of the derived indices were reported according to their currently used cutoffs in the pediatric population. Nevertheless, all values were also presented as centiles or standard deviations. Hence, BMI values between the 85th centile and the 95th centile were used to define overweight children, while any BMI value above the 95th centile was used to describe childhood obesity [8]. The same cutoffs were used for TMI as follows: values above the 85th centile were used to define overweight children, while any value above the 95th centile defined obesity [10].

Regarding VAI scores, any value above 1.58 was considered to be highly suggestive of metabolic syndrome in an indirect manner, according to the cutoff criteria in a pediatric population established by Ejtahed et al. [13].

As per the waist-to-height ratio, all values higher than 50% were considered to be proof of visceral adiposity [14].

### 2.3. Imagistic Evaluation

All subjects underwent a 2D ultrasonographic abdominal evaluation in order to establish the presence of liver steatosis. The assessment was performed by the same ultrasonographist in order to minimize inter-assay variation. A Toshiba Aplio 300 ultrasonography machine was used. Steatosis was considered whenever increased liver echogenicity was observed when compared to the right kidney parenchyma.

### 2.4. Laboratory Analysis

Blood was drawn after night fasting from every subject at admission. Part of the samples was frozen at −80 degrees Celsius, while the rest were immediately analyzed in order to assess lipid profile status, glycemic status, inflammatory status and liver function. Derived atherogenic indexes (triglyceride-to-HDL cholesterol ratio, non-HDL cholesterol, AIP—Atherogenic Index of Plasma) and HOMA-IR (homeostatic model assessment insulin resistance) values were also recorded in all enrolled patients. AIP values were used to stratify risk of atherosclerosis in three groups as follows: low risk if the AIP value was <0.1, moderate risk for AIP values of 0.1–0.24 and a high-risk group for AIP values above 0.24 [13]. HOMA-IR values above 2.5 were considered to be representative for insulin resistance [2]. Using all data provided by clinical assessment and lab results, we established the presence/absence of metabolic syndrome (MetS) according to IDF criteria in each patient [15]. Also, every obese subject enrolled was labelled either as metabolically healthy obese (MHO) or metabolically unhealthy obese (MUHO) [1].

Osteonectin was evaluated from serum samples using a commercial ELISA kit (RayBiotech USA, code ELH-SPARC1) according to manufacturer instructions. Optical densities were read using a plate reader at 450 nm (DynaRead, Bustehrad, Czech Republic). Samples were analyzed in duplicate. Limit of detection was 0.11 ng/mL, with an inter-assay coefficient of variation of 12% and intra-assay coefficient of variation of 10%.

Serum Hsp27 was evaluated using a commercial ELISA kit (RayBiotech USA, code ELH-Hsp27). Optical densities were read using a plate reader at 450 nm (DynaRead, Czech Republic). Samples were analyzed in duplicate. Limit of detection was 0.15 ng/mL, with an inter-assay coefficient of variation of 12% and intra-assay coefficient of variation of 10%.

### 2.5. NAFLD/MAFLD/PeFLD Type 2 Diagnosis

As diagnostic criteria for NAFLD have changed since 2020, in order to emphasize the importance of metabolic dysregulation in fatty liver disease (MAFLD, ex-NAFLD), we reconsidered the diagnosis of NAFLD in the initial MR PONy database according to new criteria. So, if we had initially labeled each subject who had either ultrasonographic liver steatosis and/or twice the normal level of ALT for age and gender as a NAFLD patient, we now considered all subjects with evidence of liver steatosis (either ultrasonographic, biomarker-determined or based on liver histology) and T2DM or obesity as having a MAFLD diagnosis by applying the new criteria. For the non-obese subjects, we considered the MAFLD diagnosis to be positive if they had metabolic dysregulation and liver steatosis. Metabolic dysregulation was considered when two of the following criteria were met: waist circumference above the 90th percentile for age and gender; blood pressure value above the 95th percentile for age, height and gender; serum triglyceride value above 150 mg/dL; serum HDL cholesterol value below 40 mg/dL; pre-diabetes or HOMA-IR above 2.5; and hsCRP > 2 mg/L [2]. As MAFLD diagnostic criteria were proposed and validated in the adult population, one can assume that their utility in the child population is questionable. To further address this possible issue, we mention that MAFLD diagnosis as considered in our study population is equivalent to PeFLD (Pediatric Fatty Liver Disease) type 2. PeFLD is a group of disorders recently organized under this terminology which encompasses liver steatosis due to an inherited error of metabolism (type 1), due to metabolic dysfunction (type 2 or MAFLD) or due to any (yet) unknown cause of fatty liver (type 3) [2,16].

### 2.6. Statistical Analysis

IBM SPSS 28.0.0 Statistics for Mac was used for data analysis. All normal distributed data were presented as mean and standard deviation, while non-normal distributed data were presented as median and minimum–maximum values. All categorical variables were reported as percentages. Non-parametric tests (Mann–Whitney U tests) were used to analyze differences between groups for continuous non-normal distributed data. ROC curves and AUROC calculations alongside 95% confidence intervals were computed in order to evaluate the diagnostic capacities of various variables for MAFLD diagnosis. Chi-square tests were used for analyzing differences in the frequencies of categorical variables. Statistical significance was considered for a *p* value < 0.05.

## 3. Results

A total of 71 overweight and obese patients were included in the study (42 without MAFLD and 29 with MAFLD resulting in a 40.8% prevalence of MAFLD in our study population). Baseline characteristics are presented in Table 1.

Patients with MAFLD showed higher serum Osteonectin levels compared to those of non-MAFLD patients (Figure 1, median 1692.8 ng/mL (1172.8–2435.2) vs. median 1455.2 ng/mL (635.2 ng/mL–4674.4 ng/mL, *p* = 0.003, Mann–Whitney U test)).

No difference was found in the case of serum Hsp27, although patients with MAFLD had a tendency for lower values (Figure 2, median 39.54 ng/mL (0–773.07) vs. median 41.65 ng/mL (0–583.55), *p* = 0.52, Mann–Whitney U test).

MAFLD diagnostic discrimination of BMI and TMI were assessed analyzing the resulting ROC curves. Comparable diagnostic capacity was noted as follows: TMI AUROC = 0.609 95% CI (0.47–0.74) and BMI AUROC = 0.706 95%CI (0.57–0.84)—Figure 3.

Also, a comparison between the diagnostic capacities of serum Osteonectin (SPARC), the TG/HDL ratio, AIP and VAI for MAFLD in overweight and obese children were evaluated. Analyzing the ROC curves identified a similar diagnostic capacity for the TG/HDL ratio, AIP and VAI (AUROC being 0.77 for all with a 95% confidence interval (0.59–0.84)). Serum Osteonectin had a slightly lower diagnostic capacity (AUROC 0.72 (95% CI 0.59–0.84)), as shown in Figure 4.

The TG/HDL ratio, AIP and VAI were also evaluated in the subgroup of overweight children (10 patients, 3 with MAFLD and 7 without MAFLD). Overweight patients with MAFLD also had significantly higher AIP, VAI and TG/HDL ratios compared to non-MAFLD overweight children, despite a very small sample size (Table 2). No significant differences were noted for serum Osteonectin and Hsp27 levels in this particular subgroup.

VAI and HOMA-IR were positively correlated (r = 0.36, *p* = 0.003, Spearman’s rho test).

Associations between serum Osteonectin and Hsp27 levels and the clinical and paraclinical features of the included patients were evaluated. Higher Osteonectin levels were noted in children with increased LDL cholesterol (median of 1692.80 ng/mL (1172–4674.40) vs. 1485.60 ng/mL (635.20–2435.20), *p* = 0.017, Mann–Whitney U test).

In addition, higher serum Osteonectin levels were associated with higher HOMA-IR values (above 2.50). Median serum Osteonectin in high-HOMA-IR patients was 1713.60 ng/mL (1172.80–4674.40) compared to 1476.58 ng/mL (635.70–2208.80) in patients with low HOMA-IR, *p* = 0.01, Mann–Whitney U test).

No associations with the clinical characteristics of MAFLD patients were observed for Hsp27.

## 4. Discussion

When defining obesity, the main focus is on the amount of adiposity. It is well known that the adipose tissue acts like an active metabolic endocrine organ, secreting a multitude of chemical regulators either locally or into the blood stream. Among these chemical regulators are the adipokines—cytokines that can balance or alter different metabolic chains based on their circulating level [17]. Glucose and lipid pathways, the two most important ones involved in body energy production and storage, are balanced by adipokines. When altered, insulin resistance occurrence becomes the key player in many metabolic dysregulations observed for obese subjects like hyperlipemia and hyperglycemia just to name two of them in an extremely simplified manner. Long-term metabolic dysregulation is highly associated with a myriad of important complications, including those that can impact cardiovascular system well-being. Moreover, one of the metabolic-related organ impairments (liver steatosis or MAFLD) is independently associated with cardiovascular risk as per [18,19]. Hence, many studies have focused on finding ever more accurate, minimum-invasive, and economically efficient biomarkers to highlight cardio-metabolic risk signatures in both MAFLD and non-MAFLD overweight and obese patients. Such cardiometabolic surrogate markers validated in both the adult and pediatric populations are the those of the body-fat distribution indices (BMI, TMI, VAI) [20,21] and lipid-derived indices (VAI, AIP, TG/HDL ratio) [22,23].

In this study we evaluated two biomarkers associated in the adult population with cardiometabolic risk (Osteonectin and Hsp27) in order to establish their screening diagnostic capacity for MAFLD when assessed in overweight and obese pediatric subjects. In addition, we analyzed four indices used as cardiometabolic surrogate markers in children, which are either linked to body composition or being lipid-derived, with regard to MAFLD diagnosis.

Osteonectin (SPARC) is an adipokine that exerts profibrotic properties [3], being an extracellular matrix component that is implied in adipose tissue fibrosis development. Once the adipose tissue becomes fibrotic, it can no longer accumulate triglycerides that are now deposited in ectopic sites like hepatic cells (MAFLD development). As such, high Osteonectin expression is also associated with high levels of circulatory triglycerides and possibly MAFLD [4].

Data on serum Osteonectin in pediatric population is scarce in the literature. There is only one previously published study that evaluated serum Osteonectin alongside other adipokines in a cohort of Chinese obese patients [24]. Serum Osteonectin levels were higher in obese patients that also expressed insulin resistance. Unfortunately, in that study, there was no hepatic evaluation for MAFLD used for the patients.

We confirmed Osteonectin as a biomarker for pediatric MAFLD as we found that circulatory Osteonectin levels were higher in pediatric MAFLD patients compared to non-MAFLD pediatric patients. These findings are similar to previous studies on adults and “in vivo” studies [4]. It was previously suggested that Osteonectin could be a stronger predictor for liver fibrosis in MAFLD patients compared to cytokeratin 18 [4]. Also, Osteonectin was associated with the expression of profibrotic markers like TGF-beta 1, and a lower expression was associated with a lower risk for NASH development in rodent models [4]. Like previous studies, we found that patients with higher insulin resistance expressed higher Osteonectin levels.

In addition, we found that patients with MAFLD had a higher TG/HDL ratio, a feature that is also present in the overweight patient subgroup despite the fact that there was no difference in serum Osteonectin levels compared to those of the healthy controls in this particular subgroup. We also found that Osteonectin levels were significantly higher in all subjects who had a high LDL cholesterol level, regardless of their MAFLD status or their inclusion in the overweight or obese (BMI-based) group. This finding indicates Osteonectin as an important biomarker for obesity-related risk stratification as it may be significantly expressed only in an obesogenic milieu or profoundly dyslipidemic one, while the TG/HDL ratio seems to be an accurate marker for organ impairment associated with metabolic dysregulation even before obesity is developed. As such, one can speculate about Osteonectin even being a sensitive marker for lipid pathway dysregulation that might serve as signature of future obesity-induced atherosclerotic and steatosis-related complications.

The other molecule we evaluated in overweight and obese pediatric subjects is Hsp27. Hsp 27 is a heat shock protein and is part of the small Hsp (sHsp) family of proteins with a low molecular weight, which are known as chaperons. Hsp 27 is expressed in many normal tissues, including the heart, skin, lung, uterus, cervix, placenta and also in some human tumoral cells like those derived from the breast and prostate [25]. When cells are under any kind of stress, Hsp27 suffers a process of phosphorylation by reorganizing itself into smaller tetramers and dimers in a dynamic manner induced by cellular conditions. With this reorganized conformation, Hsp27 can interact with other proteins where it can facilitate refolding them and can interact with cell migration by modulating F-actin through F-actin polymerization inhibition [25]. Hsp27 also has an anti-oxidative role alongside its interaction with the proteosome degradation machinery; nevertheless, it has an anti-apoptotic role [26]. Briefly, the production of Hsp27 in stress conditions is cytoprotective, and it is currently thought that it has an important role after cardiac ischemia or myocardic injury [25]. In our study, we found a tendency for MAFLD pediatric subjects (either overweight or obese) to have lower Hsp27 levels than those of the non-MAFLD subjects, although differences were not statistically significant, most probably due to low statistical power (small number of subjects). Nevertheless, this tendency speaks volumes about the same aspect already highlighted in “in vitro” studies, like the one published in 2016 by Sookoian S et al. [27] which states that Hsp27 is downregulated in the ballooning degeneration of hepatocytes (a characteristic feature associated with MAFLD progression). The possible explanation suggested by Sookoian S et al. resides in the impossibility of the ballooned hepatocytes to properly react to metabolic-induced stress [27]. As far as we know, our study is the first one to question Hsp27 responses to metabolic challenges in the pediatric population. While data on Hsp27 behavior in the pediatric population are insufficient, this tendency found in our study might come to be an important clue to further studies.

Subsequently, another issue we had in mind when analyzing the gathered data from the MR PONy cohort was that of the relevance of body composition indices being metabolic disfunction markers as they are a measure of metabolically active adiposity distribution. Mainly, we emphasized on TMI and BMI role as a possible predictor for MAFLD diagnosis in overweight and obese children. TMI (Triponderal Mass Index) is a novel index used to better appreciate adiposity in children and adolescents. It is well known to all pediatricians and pediatric endocrinologists that BMI is an imperfect tool to screen body fat composition. As BMI is an accurate tool to predict obesity in adults, it fails to do the same in adolescent and pediatric populations where there is evidence that weight does not scale with squared height [10].

Of course, BMI Z-scores are useful for classifying children and adolescents as lean, overweight or obese, but they also have limitations imposed by the ever-changing body proportions and fat composition of developing children. As Cole et al. reported how the true scaling powers under the age of 18 years are somewhere between 2.5 and 3.5 [28], TMI is definitely a more adequate tool to be used when evaluating and reporting childhood obesity than BMI. Nevertheless, TMI is stable during adolescence until 20 years [10]. Moreover, when compared to BMI, TMI is proven to be a better predictor of metabolic syndrome, including cardiometabolic risk, in children and adolescents [29]. Nevertheless, TMI values are cited to be intimately correlated with insulin resistance, the key component of metabolically unhealthy obesity [30]. Having all this in mind, we questioned TMI behavior in relation to MAFLD in obese children. In our study, TMI values for non-MAFLD pediatric subjects were significantly lower than those of their peers diagnosed with metabolically associated fatty liver disease (MAFLD), which are similar to the findings of a recent paper published by Basarir G et al. [31]. Meanwhile, BMI values were not statistically significant different between the two subject groups (non-MAFLD versus MAFLD).

In terms of cardio-metabolic health, body fat distribution indices are not the only ones recognized as surrogate markers. The composite lipid-derived indices are a measure of obesity-derived atherogenic lipids [32]. Three well-known lipid-derived indices are VAI, AIP and the TG/HDL ratio, which are confirmed as cardio-metabolic risk factors in both pediatric and adult populations [32,33].

In our study, we found that the diagnostic capacities of the TG/HDL ratio, AIP and VAI for MAFLD in overweight and obese children were worthy, as TG/HDL AUROC was 0.773 with a CI of 95% (0.656–0.891) almost identical to AIP AUROC which was also 0.773 with a CI of 95% (0.656–0.890) and very similar to VAI AUROC (0.770 with a CI of 95% (0.648–0.891)) with *p* < 0.000. Interestingly, when comparing these findings to the already known body of literature, we concluded that lipid-derived indices have a better discrimination capacity for MAFLD diagnosis in children than the currently used screening tools of ultrasonography and ALT level. Draijer LG et al. proved in 2019 that screening the test accuracy used for metabolic dysfunction-associated fatty liver disease in children is comparable when referring to ALT and ultrasonography as follows: 0.74 (95% CI 0.65–0.83) and 0.70 (95% CI 0.60–0.79), respectively (*p* = 0.41) [34]. Accordingly, we strongly suggest considering each child who has abnormal lipid-derived indices as already having metabolic organ impairment and managing them accordingly in a timely manner.

## 5. Conclusions

To summarize, the main findings of this study are as follows: (1) Confirmation of Osteonectin as a diagnostic biomarker for pediatric MAFLD and as an important biomarker for obesity-related risk stratification as it may be significantly expressed only in an obesogenic milieu or profoundly dyslipidemic one, while the TG/HDL ratio seems to be an accurate marker for organ impairment associated with metabolic dysregulation even before obesity has developed. (2) The association between serum Osteonectin and hyperlipidemia (high LDL cholesterol) in pediatric subjects. (3) The positive correlation between the surrogate insulin resistance marker (HOMA-IR) and the Visceral Adiposity Index in the pediatric population. (4) The better diagnostic discrimination capacity than currently used screening tools (AST and ultrasonography) of lipid-derived indices (TG/HDL ratio, AIP, VAI) for MAFLD diagnosis in pediatric patients, which concludes with our strong advice to consider each child who has abnormal lipid-derived indices as already having metabolic organ impairment and to manage them accordingly in a timely manner. (5) Nevertheless, we emphasize the tendency for MAFLD pediatric subjects (either overweight or obese) to have lower Hsp27 levels than those of non-MAFLD subjects, which might be an indirect measure of an abnormal (metabolic) stress response in affected hepatocytes.

These findings come to complete the bigger picture of pediatric metabolic dysfunction-associated liver disease as pediatricians, endocrinologists and public health experts are all struggling to address MAFLD in times when current interventions to flatten the curb of childhood and adolescent obesity seem to have failed.

In the end, we want to highlight that our findings have some limitations related to our study design (cross-sectional approach) and to the small number of subjects. Nevertheless, a main limitation comes from the lack of a histologic diagnosis of MAFLD in our cohort as we used only ultrasonography and the ALT level in that scope. Another limitation of the study is due to the fact that we have not considered the metabolic risk assessment apolipoproteins as it was beyond the scope of the study. Similarly, we used HOMA-IR as a surrogate marker for insulin resistance as it is universally accepted even if it is not always accurate. However, despite these limitations, the findings reported here can have a high impact on the specific field of obesity-related organ impairment in the pediatric population, needing a further superior study approach.

## Figures and Tables

**Figure 1 nutrients-16-00866-f001:**
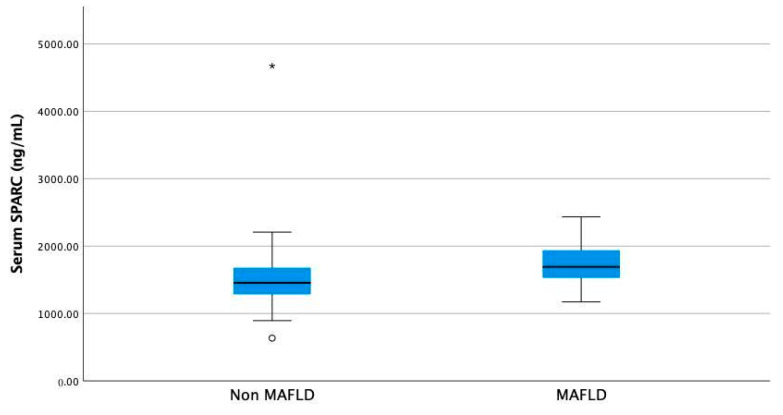
Serum Osteonectin (SPARC) levels in non-MAFLD compared to MAFLD overweight and obese children. Differences are statistically significant, with *p* = 0.003 (Mann–Whitney U test). *—outliers.

**Figure 2 nutrients-16-00866-f002:**
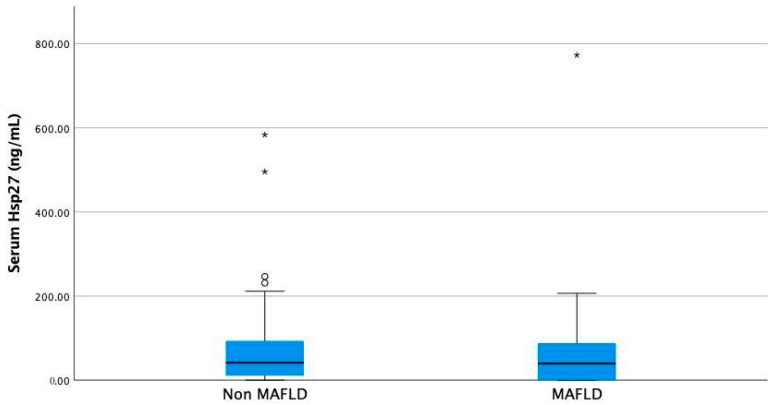
Serum Hsp27 levels in non-MAFLD compared to MAFLD overweight and obese children Differences are not statistically significant, with *p* = 0.52 (Mann–Whitney U test). *—outliers.

**Figure 3 nutrients-16-00866-f003:**
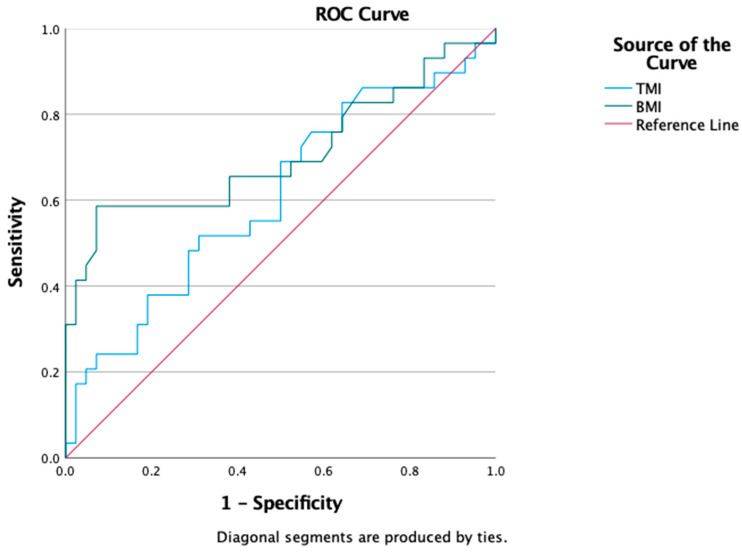
ROC curves of TMI and BMI for MAFLD diagnosis.

**Figure 4 nutrients-16-00866-f004:**
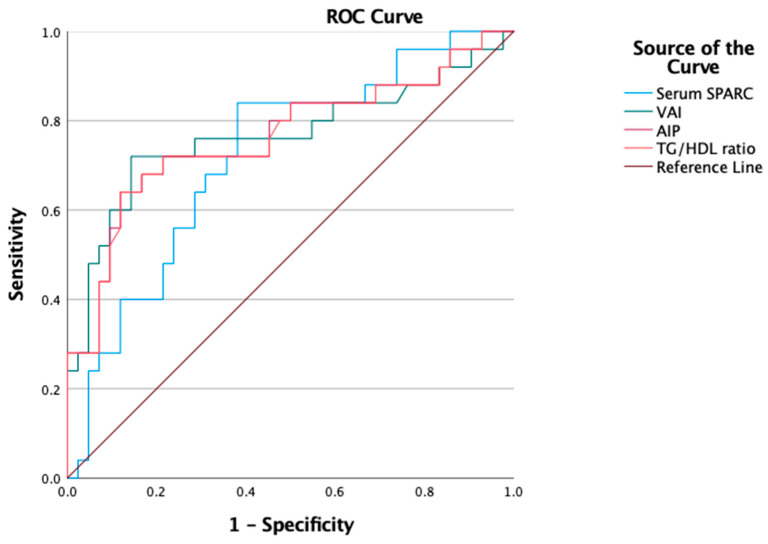
ROC curves of serum SPARC, TG/HDL ratio, AIP and VAI for MAFLD diagnosis in overweight and obese children. AUROC for serum SPARC being 0.72 with a 95% CI (0.59–0.84), AUROC for the TG/HDL ratio being 0.77 with a 95% CI (0.65–0.89), AUROC for VAI being 0.77 with a 95% CI (0.65–0.89), AUROC for AIP being 0.77 with a 95% CI (0.65–0.89).

**Table 1 nutrients-16-00866-t001:** Descriptive data of the 71 patients included.

Variable	Non-MAFLD Patients (n = 42)	MAFLD Patients (n = 29)	*p*-Value
Male gender (n, %)	23 (54.8%)	14 (48.3%)	0.63
**Age (years)**	**8.50 (3–16)**	**12 (6–17)**	**0.007**
MUHO (n, %)	33 (78.6%)	24 (82.8%)	0.66
**WC (percentile)**	**95 (75–99)**	**95 (90–99)**	**0.048**
MAC (percentile)	95 (75–99)	95 (25–99)	0.95
WtHR	58.60 (45.56–68.70)	61.40 (50–78.28)	0.058
**BMI (kg/m^2^)**	**24.04 (18.40–32.70)**	**29 (17.60–40.43)**	**0.003**
BMI percentile	98 (88–99)	98 (85–99)	0.62
BMI z-score	2.00 (1.18–4.00)	2.08 (1.04–3.14)	0.51
**TMI**	**16.96 (14.19–24.91)**	**18.24 (13.45–26.48)**	**<0.001**
TMI (percentile)	99 (75–99)	99 (50–99)	0.09
HTA (n,%)	18 (42.9%)	16 (55.2%)	0.34
Dyslipidemia (n,%)	28 (66.7%)	22 (75.9%)	0.44
**TG (mg/dL)**	**71 (30–195)**	**121 (44–251)**	**<0.001**
**HDL cholesterol (mg/dL)**	**45 (33–70)**	**37 (19–68)**	**0.008**
LDL cholesterol (mg/dL)	98 (53–156)	109 (51–156)	0.38
**Non-HDL cholesterol (mg/dL)**	**114.5 (50–170)**	**128 (62–188)**	**0.22**
**TG/HDL ratio**	**1.76 (0.52–5.00)**	**3.49 (0.81–8.58)**	**<0.001**
**Insulin (mU/L)**	**7.71 (0.96–41.41)**	**16.90 (2.05–41.57)**	**0.015**
**HOMA-IR**	**1.67 (0.34–13.91)**	**3.69 (0.55–10.57)**	**0.006**
**VAI**	**2.50 (0.59–10.31)**	**5.45 (0.92–16.35)**	**<0.001**
**AIP**	**–0.11 (–0.63–+0.64)**	**0.18 (–0.45–+0.57)**	**<0.001**
**AIP group stratification (n,%)**			
**1 (low risk)**	**34 (81%)**	**11 (37.9%)**	**<0.001**
**2 (intermediate risk)**	**25 (11.9%)**	**6 (20.7%)**	
**3 (high risk)**	**3 (7.1%)**	**12 (41.4%)**	

WC: waist circumference, MAC: midarm circumference, BMI: Body Mass Index, TMI: Triponderal Mass Index, HTA: hypertension, TG: triglycerides, TG/HDL ratio: triglyceride-to-HDL ratio, WtHR: waste-to-height ratio, NLR: neutrophil-to-lymphocyte ratio, HOMA-IR: Homeostatic Model Assessment of Insulin Resistance, AIP: atherosclerotic index of plasma VAI: visceral adiposity index. *p*-values were determined using Mann–Whitney tests for continuous variables and Fisher’s exact test for nominal variables. Continuous variables are presented as median, minimum and maximum values. Bolded variables were found to be significant (*p* < 0.05).

**Table 2 nutrients-16-00866-t002:** AIP, VAI and TG/HDL ratio in overweight non-MAFLD and MAFLD patients.

	Overweight Non-MAFLD (n = 7)	Overweight MAFLD (n = 3)	*p*-Value
AIP	−0.13 (−0.51–+0.02)	0.25 (0.08–0.57)	0.017
VAI	2.23 (1.46–5.03)	5.45 (5.24–16.35)	0.017
TG/HDL ratio	1.69 (0.71–2.37)	4.14 (2.73–8.58)	0.017

TG/HDL ratio: triglyceride-to-HDL ratio, AIP: atherosclerotic index of plasma, VAI: visceral adiposity index.

## Data Availability

Data used to support the findings of the present study are available upon request from the corresponding author.

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
