# Peer review of "Exploring the Role of Serum Osteonectin and Hsp27 in Pediatric MAFLD Diagnosis and Cardiometabolic Health"

_nutrients, 2024, doi:10.3390/nu16060866_

Round 1

Reviewer 1 Report

Comments and Suggestions for Authors

The research sought to assess the usefulness of the adipokine osteonectin (SPARC) and the heat-shock protein Hsp27 as indicators for diagnosing type 2 pediatric fatty liver disease (MAFLD) and to examine their connections with the physical and biochemical characteristics of overweight and obese children. This study appears to be the first to compare these biomarkers within the pediatric MAFLD spectrum. Additionally, the study aimed to investigate three cardio-metabolic risk indices (TG/HDL ratio, VAI, and AIP) as potential indicators for identifying MAFLD.

This article is excellently crafted and captivating. Both the title and abstract succinctly convey the essence of the research, offering ample and pertinent details. The introduction provides relevant background information, while the methods are clearly elucidated, ensuring transparency and reproducibility. The authors present data consistent with the described methodology, complemented by relevant tables. The article follows a logical organization and is readily understandable. The conclusions are well-aligned with the presented evidence and arguments. Furthermore, the references are up-to-date and comprehensive.

The only my remarks:

1.please the limitations of the study

Author Response

We thank the reviewer kind comments and appreciations. We have considered adding more limitations of the study. 

Reviewer 2 Report

Comments and Suggestions for Authors

This great article examines biomarkers for steatosis in children. The setup has some flaws and the data are succinct, but most of this in inherent to studies in pediatrics. I have some comments:

- The entity is mostly known as Metabolic dysfunction–associated steatotic liver disease (MASLD).

- ALT is used for diagnosis, but AST may be a better marker.

- The ponderal index is used as a better alternative for BMI. However, both the body adiposity index (BAI) and the body shape index (BSI) may be more evidence-based.

- To evaluate metabolic risk, both apoB ad apoE might be a better alternative to TG and HDL.

- The HOMA-index is not always reliable to estimate insulin resistance.

Comments on the Quality of English Language

Some sentences need to be re-structured.

Author Response

We thank the reviewer for the comments and appreciatons.

  1. As we are aware of the new ethimology of this entity (former NAFLD) we decided to keep the MAFLD name as it is the one suggested by the International Expert Consensus statement referring to pediatric population published in Oct 2021 by ESLAM et al. DOI: 10.1016/S2468-1253(21)00183-7
  2. We used ALT as this is the current protocol of assessment for pediatric NAFLD/MAFLD screening recommended by ESPHGAN DOI: 10.1016/j.jhep.2024.01.004) and NASPGHAN (DOI: 10.1097/MPG.0000000000001482) 
  3. BAI overestimates fat in children. More appropriate is pediatric BAI score (pBAI). However we did not measure hip circumference in our study population hence we did not use this particular body composition index [El Aarbaoui T, Samouda H, Zitouni D, di Pompeo C, de Beaufort C, Trincaretto F, Mormentyn A, Hubert H, Lemdani M, Guinhouya BC. Does the body adiposity index (BAI) apply to paediatric populations? Ann Hum Biol. 2013 Sep-Oct;40(5):451-8. doi: 10.3109/03014460.2013.802011. Epub 2013 Jun 19. PMID: 23777297.]. Also, ABSI analysis was beyond the scope of the study, as it was also previously demonstrated that it has a similar prognostic value as BMI (at least for hypertension) Ge W, Yi L, Xiao C, Xiao Y, Liu J, Liang F, Yin J, Hu J. Effectiveness of a body shape index in predicting pediatric high blood pressure. Pediatr Res. 2022 Sep;92(3):871-879. doi: 10.1038/s41390-021-01844-5. Epub 2021 Nov 16. PMID: 34785781.

    •  
  4.  We acknowledge in the limitations of the study that these markers were not evaluated.
  5. We acknowledge in the limitations of the study that HOMA-IR is not necessarily reliable tool for insulin resistance assessment.
  6. Some phrases were reconsidered and restructured and also some minor spelling errors were corrected. 

Reviewer 3 Report

Comments and Suggestions for Authors

This manuscript is written well; however this will be able to revise a few points.

1) Authors described that " To date, there is little data in literature 58 addressing Hsp27 role in MAFLD development and to the best of our knowledge no re- 59 port about Hsp27 and PeFLD type 2 patients." However, we do not understand this study's hypothesis in introduction. So, could you show this more clearly? 

2) Authors described that ", we gathered and (re)analyzed data from MR PONy 71 (Metabolic and cardiovascular Risk factors in a Pediatric Overweight and obese popula- 72 tion with or without NAFLD) cohort (already described in our previously published re- 73 search papers. " in the methods. However, we do not know these methods of selection in the subjects. So, could you show this using flow chart as a Figure?

3) Authors showed that " All normal distributed 177 data were presented as mean and standard deviation, while non-normal distributed data 178 were presented as median and minimum-maximum values. All categorical variables were 179 reported as percentages. Non-parametric tests were used to analyze differences between 180 groups. ROC curves were used for the evaluation of diagnostic capacities of various vari- 181 ables for MAFLD diagnosis. Statistical significance was considered for p value < 0.05." this is very simple methods in the present study, so we will not be able to understand these methods. So, could you re-write these methods more exactly? 

4) Figures are not clearly for readers, so it is not easy to understand, so could you change these more clearly? 

5) Have you considered any hypotheses regarding the notation of results?

6)Are there any other limitations to this study?

Comments on the Quality of English Language

N/A

Author Response

We thank the reviewer for his comments.

1. We have rephrased the aim of the study as following " Therefore, the aim of the study was to evaluate newly candidate biomarkers adipokine osteonectin (SPARC) and heat-shock protein Hsp27 for PeFLD type 2 (MAFLD) diagnosis and risk stratification. In addition, we aimed to assess the association of their circulatory expression with auxological and biochemical profiles of overweight and obese pediatric subjects". We have deleted "To date, there is little data in literature addressing Hsp27 role in MAFLD development and to the best of our knowledge no report about Hsp27 and PeFLD type 2 patients"

2. The method of selection involves recruitment of each admitted child (consecutive patients) in our hospital in the mentioned timeline (jan-dec 2017) which gathered the inclusion criteria mentioned and if the parents or legal guardian signed informed consent and agreed to  participate in the study. 

3.We have expanded the Statistical Approach section as follows: "All categorical variables were reported as percentages. Non-parametric tests (Mann-Whitney U tests)  were used to analyze differences between groups for continuous non-normal distributed data. ROC curves and AUROC calculations alongside with 95% confidence intervals were computed in order to evaluate the diagnostic capacities of various variables for MAFLD diagnosis. Chi-square tests were used for analyzing differences in the frequencies of categorical variables. Statistical significance was considered for a p value < 0.05."

4. We have changed figure 3 and figure 4, adding all ROC curves in the same graph.

5. We are not sure that we understand what the reviewer tried to ask us, but we have tried in the manuscript to find several associations between the molecular function of the biomarkers and clinical associations that were found. We tried not to be overspeculative with other hypothesis since this was only a cross-sectional study. 

6. We have added more limitations of the study at the end of the manuscript.